# Identification of a Stable, Non-Canonically Regulated Nrf2 Form in Lung Cancer Cells

**DOI:** 10.3390/antiox10050786

**Published:** 2021-05-15

**Authors:** Sara Mikac, Michał Rychłowski, Alicja Dziadosz, Alicja Szabelska-Beresewicz, Robin Fahraeus, Theodore Hupp, Alicja Sznarkowska

**Affiliations:** 1International Centre for Cancer Vaccine Science, University of Gdansk, Kladki 24, 80-822 Gdansk, Poland; sara.mikac@phdstud.ug.edu.pl (S.M.); alicja.dziadosz@ug.edu.pl (A.D.); robin.fahraeus@inserm.fr (R.F.); ted.hupp@ed.ac.uk (T.H.); 2Laboratory of Virus Molecular Biology, Intercollegiate Faculty of Biotechnology, University of Gdansk, Abrahama 58, 80-307 Gdansk, Poland; michal.rychlowski@biotech.ug.edu.pl; 3Department of Mathematical and Statistical Methods, Poznań University of Life Sciences, 28 Wojska Polskiego St, 60-637 Poznań, Poland; aszab@up.poznan.pl; 4Inserm UMRS1131, Institut de Génétique Moléculaire, Université Paris 7, Hôpital St. Louis, F-75010 Paris, France; 5RECAMO, Masaryk Memorial Cancer Institute, Zluty kopec 7, 65653 Brno, Czech Republic; 6Department of Medical Biosciences, Building 6M, Umeå University, 901 85 Umeå, Sweden; 7Edinburgh Cancer Research Centre at the Institute of Genetics and Molecular Medicine, Edinburgh University, Edinburgh EH14 1DJ, UK

**Keywords:** Nrf2 detection, Nrf2 antibodies, Nrf2 migration in SDS-PAGE

## Abstract

Nrf2 (nuclear factor erythroid 2 (NF-E2)-related factor 2) transcription factor is recognized for its pro-survival and cell protective role upon exposure to oxidative, chemical, or metabolic stresses. Nrf2 controls a number of cellular processes such as proliferation, differentiation, apoptosis, autophagy, lipid synthesis, and metabolism and glucose metabolism and is a target of activation in chronic diseases like diabetes, neurodegenerative, and inflammatory diseases. The dark side of Nrf2 is revealed when its regulation is imbalanced (e.g., via oncogene activation or mutations) and under such conditions constitutively active Nrf2 promotes cancerogenesis, metastasis, and radio- and chemoresistance. When there is no stress, Nrf2 is instantly degraded via Keap1-Cullin 3 (Cul3) pathway but despite this, cells exhibit a basal activation of Nrf2 target genes. It is yet not clear how Nrf2 maintains the expression of its targets under homeostatic conditions. Here, we found a stable 105 kDa Nrf2 form that is resistant to Keap1-Cul3-mediated degradation and translocates to the nucleus of lung cancer cells. RNA-Seq analysis indicate that it might originate from the exon 2 or exon 3-truncated transcripts. This stable 105 kDa Nrf2 form might help explain the constitutive activity of Nrf2 under normal cellular conditions.

## 1. Introduction

Transcription factor Nrf2 was identified in 1994 as a protein that binds to the same tandem repeat of consensus DNA sequence in the promoter of beta-globin gene as activating protein 1 (AP-1) and the nuclear factor erythroid 2 (NF-E2) and contains a conserved basic leucine zipper (bZIP) DNA binding domain highly homologous to that of NF-E2 [1]. It belongs to the Cap ‘N’ Collar (CNC) family that contains a conserved basic leucine zipper (bZIP) structure. Nrf2 is considered one of the major regulators of cellular defense and survival and activates cellular antioxidant response by inducing the transcription of a wide array of genes that are responsible for the protection against extrinsic and intrinsic insults, including oxidative stress and xenobiotics [1,2]. The main Nrf2 target genes represent the most important cytoprotective defense system in cell, including genes responsible for glutathione and thioredoxin production and regeneration, NADPH regeneration, heme and iron metabolism, reactive oxygen species (ROS), and xenobiotic detoxification [3,4].

The activation and inactivation of the Nrf2 pathway is primarily regulated by Keap1 (Kelch-like erythroid cell-derived protein with CNC homology [ECH]-associated protein 1), a substrate adaptor for a Cul3-containing E3 ubiquitin ligase [5]. Under basal conditions, Keap1-Cul3-E3 ligase complex is activated, causing the ubiquitination and degradation of Nrf2 [6]. In response to extrinsic and/or intrinsic insults, Keap1-dependent ubiquitin ligase activity is inhibited and the Nrf2 protein is accumulated. It leads to the translocation of Nrf2 to the nucleus and activation of the transcription of its target genes [4,6]. Since under homeostatic conditions the Nrf2 is thought to be constitutively degraded, it is not clear how the basal expression of its target genes is maintained. 

Although activation of Nrf2 has a protective role against various toxicants and diseases, the prolonged activation has been shown to favor a progression of several types of cancers, such as lung, breast, head and neck, ovarian, and endometrial carcinomas [5,7,8,9]. There are several mechanisms by which Nrf2 signaling pathway is constitutively activated in cancer cells: (1) somatic mutations in Keap1 or the Keap1 binding domain, that disrupt binding of Nrf2 and Keap1; (2) epigenetic silencing of Keap1; (3) accumulation of disruptor proteins such as p62, which leads to the dissociation of the Nrf2-Keap1 complex; (4) transcriptional induction of Nrf2 by oncogenic K-Ras, B-Raf and, c-Myc; and (5) post-translational modification of Keap1 cysteines by succinylation [4,5,6,7,8,9,10,11]. Nrf2 also has an effect on metabolic reprogramming, redirecting glucose and glutamine to synthesis pathways of purine nucleotides, glutathione, and serine [8]. Therefore, it enhances aggressive cancer cell proliferation and promotes chemoresistance and radioresistance.

Most often studies on Nrf2 include antibodies, but there are concerns regarding Nrf2 migration in SDS-PAGE and the specificity of some anti-Nrf2 antibodies [10,11]. Nrf2 is phosphorylated by various kinases, but even after phosphatase treatment two Nrf2 forms were detected [12,13,14]. The origin of these forms is not clear. Latest studies indicate that alternative splicing in lung and head and neck cancers produces Nrf2 forms of increased stability that lack exon 2 or exon 2 and 3 [15].

In this work we evaluated the specificity of two commercial antibodies, Abcam EP1808Y and Cell Signalling D1Z9C, in detecting Nrf2 forms in cells with different Nrf2 activation status and found that EP1808Y antibodies recognize extremely stable, non-canonically regulated 105 kDa Nrf2 form that translocates to the nucleus and, most probably, originates from the alternatively spliced *NFE2L2* transcripts. These results imply the regulation of Nrf2 activity by the expression of forms with different stability translocating to the nucleus and can help explain how the basal expression of Nrf2 transcription targets is maintained under physiological conditions.

## 2. Materials and Methods

### 2.1. Cell Lines 

Non-small cell lung cancer cell lines A549 and RERF-LC-AI were purchased from RIKEN BRC Cell Bank (Tsukuba, Ibaraki, Japan) and CRISPR/Cas9-induced NRF2 knockout in A549 cells (clone 2-11) was constructed and kindly provided by Prof Eric Kmiec (Gene Editing Institute, Christiana Care Health System, Newark, NJ, United States). All cell lines were cultured in Dulbecco’s modified Eagle’s medium (Gibco, Thermo Fisher Scientific), with 8% of Fetal Bovine Serum (Gibco, Thermo Fisher Scientific, Waltham, MA, United States) and 1% of Penicillin-Streptomycin (10,000 U/mL, Gibco, Thermo Fisher Scientific). Cells were maintained at 37 °C under humidified conditions with 5% CO2.

### 2.2. Lipid-Mediated Transfection

Cells were seeded in the 12-well plates, 100,000 cells/well. 24 h after seeding, cells were transfected with control siRNA-A (ON-TARGET plus^TM^ Control Pool, Dharmacon^TM^, referred in text as scrRNA), as a control for transfection (25 nM), and small-interfering RNA (siRNA, ON-TARGET plus^TM^ SMART pool, Dharmacon^TM^) in concentration of 10 and 25 nM, with 3 μL/well of Lipofectamine 3000 reagent (Invitrogen, Thermo Fisher Scientific), according to manufacturer’s instructions. Western blot was performed 48 h after transfection.

### 2.3. Western Blot Analysis

Total protein was acquired by lysing cells in RIPA buffer. Proteins were electrophoretically separated via 8% SDS-PAGE and transferred to nitrocellulose blotting membrane (Amersham Protran^®^). To block the membranes, 5% non-fat milk in Tris-buffered saline was applied at room temperature for half an hour. Membranes were subsequently incubated overnight with, anti-NRF2 (EP1808Y)—ChIP Grade (cat. no. ab62352; Abcam, Cambridge, UK), anti-NRF2 (D1Z9C) XP antibody (cat. no. 12721; Cell Signaling Technology), and anti-β-actin (cat. no. A2228; Sigma-Aldrich, St. Louis, MI, United States) in blocking buffer at 4 °C at 1:500 dilution. Subsequently, membranes were washed three times in TBST followed by incubation for 1 h with HRP-labeled goat anti-rabbit/mouse IgG (Jackson ImmunoResearch, West Grove, PA, United States) at a 1:3000 dilution and washed in TBST again. Bands were visualized using chemiluminescent substrate (Clarity Max^TM^ Western ECL Substrate, BIO-RAD, Hercules, CA, United States).

### 2.4. Immunofluorescence 

Cells were seeded on 15 mm coverslips in a 12-well plate and fixed with 4% paraformaldehyde (PFA) for 10 min, rinsed 3 times with PBS and incubated 5 min with 0.2% Triton ×100 for permeabilization. After rinsing 3 times with PBS, cells were blocked with 5% BSA in PBS, overnight at 4 °C. The next day, cells were stained with primary antibodies: anti-NRF2 [EP1808Y]—ChIP Grade (cat. no. ab62352; Abcam) and anti-NRF2 (D1Z9C) XP antibody (cat. no. 12721; Cell Signaling Technology) at 1:500 dilution, at RT for 2 h. They were washed 3 times with 1% BSA in PBS and stained with secondary antibodies (Alexa Flour 488 goat anti-rabbit; ThermoFisher Scientific; 1:2000), in the dark at RT, for 1 h, washed 3 times with 1% BSA in PBS and mounted using ProLong Diamond Antifade Mountant (ThermoFisher Scientific). Specimens were imaged using a confocal laser scanning microscope (Leica SP8X, Mannheim, Germany) with a 63× oil immersion lens.

### 2.5. Treatment with Translation Inhibitors—Cycloheximide and Emetine Dihydrochloride 

Cells were seeded in the 6-well plates, 300,000 cells/well. Forty-eight hours later cells were treated with cycloheximide (10 µM) and emetine dihydrochloride (20 µM), for 8, 16, and 24 hours. Cells were collected and analyzed by Western blot.

### 2.6. Treatment with Neddylation Inhibitor MLN4924

Cells were seeded in the 6-well plate, 500,000 cells/well. 24 h later, cells were treated with neddylation inhibitor MLN4924 (1 μM) for 12 h. Cells were collected and analyzed by Western blot. 

### 2.7. Treatment with Lambda Protein Phosphatase (λ PP)

Firstly, 800,000 cells were lysed in 250 µL of RIPA lysis buffer, sonicated for 15 min on ice and briefly centrifuged at 13,000× *g*. For dephosphorylation, 40 µL of cell lysate was incubated with 400 U of λ PP (New England Biolabs) in the dedicated buffer and in the presence of manganese ions at 30 °C for 30 min. Control samples underwent the same treatment, but without the enzyme. The Nrf2 phosphorylation was analyzed by Western blot.

### 2.8. Fractionation

Cell pellets were collected from T75 bottles, 10,000,000 cells/bottle, after trypsinization and washing with PBS. Then, 500 µL of fractionation buffer was added to the pellets and samples were incubated for 15 min on ice. Fractionation buffer contained 20 mM HEPES (pH 7.4), 10 mM KCl, 2 mM MgCl_2_, 1 mM EDTA, and 1 mM EGTA. Just before use, 1 mM DTT and PI Cocktail were added. Cells were lysed in fractionation buffer by gentle pipetting every few minutes, while kept on ice for 20 min. After that, samples were centrifuged at 3000 rpm for 5 min. The pellet contained nuclei and the supernatant contained cytoplasm. Samples were analyzed by Western blot and lamin B1 (anti-Lamin B1, PA5-19468, Thermo Fisher) was used as a nuclear fraction marker.

### 2.9. Analysis of NFEL2L2 Transcripts Expression in A549 Cells

To assess the expression of specific *NFE2L2* transcripts in A549 cell line, we have used the RNA sequencing data from the project available at NCBI Gene Expression Omnibus (GEO) public database (accession numbers GSM2308412, where A549 cell line was sequenced with paired Illumina protocol. Primary analysis of RNA-seq data included the quality control of sequenced reads with the use of FastQC (Andrews, 2010), reads trimming with the usage of Trimmomatic [16] and mapping to the reference genome based on NCBI reference human genome (assembly GRCh38.p13) and annotation (release 109) [17] with the Hisat2 aligner [18]. Further data preparations was performed with SAMtools software [19] and R software [20] together with Bioconductor platform. Assembly of RNA-Seq alignments into potential transcripts together with calculation of their expression levels were performed with StringTie software [21]. Visualization of the alignments, identified transcripts and junctions was performed in IGV software [22].

## 3. Results

In this study we have used two lung cancer cell lines: adenocarcinomic human alveolar basal epithelial cells A549 and a squamous cell carcinoma RERF-LC-AI (further referred as RERF) that differ in the Nrf2 level and activation status. A549 cells have a high steady-state level of constitutively active Nrf2 attributed to the homozygous *KEAP1* mutation (G333C) that disrupts binding with Nrf2 leading to Nrf2 accumulation and activation of its transcriptional programs [23]. Another reason for such a high Nrf2 level in these cells is the trisomy of the chromosome 2 with Nrf2 gene (while *KEAP1* is localized on the disomic chromosome 19) [24]. We have also made use of A549 Nrf2 functional KO cells which bear lower levels of Nrf2 as two out of three alleles have been successfully knocked out with CRISPR/Cas9 technology and the third allele has the ‘in frame’ deletion within the nuclear export signal (NES) thus the expressed Nrf2 cannot re-enter the nucleus [25]. The RERF cells have a wild type *KEAP1* and therefore low Nrf2 levels under no stress conditions [26].

Firstly, we analyzed the Nrf2 migratory pattern using 8% sodium dodecyl sulfate– polyacrylamide gel electrophoresis (SDS-PAGE) (Figure 1a,b) with Cell Signaling D1Z9C and Abcam EP1808Y antibodies. The epitope of D1Z9C antibodies is located in the middle of Nrf2 protein, while of EP1808Y antibodies towards the C-terminus (Figure 1c). Both antibodies detected two bands in A549 Nrf2 wt cells, one of ~105 kDa and the upper, below 130 kDa., while in RERF cells only the lower 105 kDa was detected under these conditions. Interestingly, in the functional Nrf2 KO cells, both the 105 kDa and 130 kDa signals were significantly weaker or disappeared, indicating that these bands are Nrf2-specific.

Similarly, when we knocked down the *NFE2L2* (gene encoding Nrf2) with the pool of Nrf2 targeting siRNAs (ON-TARGET plus^TM^ SMART pool, Dharmacon^TM^), we observed a decrease in the signal from tested antibodies in both A549 and RERF cell lines after 25 nM siNrf2 (Figure 2). D1Z9C antibodies detected a 130 kDa band that decreased upon 10 nM siNrf2 in A549 and upon 25 nM siNrf2 in RERF cells. A 105 kDa band was not detected under these conditions by D1Z9C antibodies. As before, EP1808Y antibodies detected two Nrf2 signals: at ~130 kDa (similar to those detected by D1Z9C), much stronger in A549 cells, and at 105 kDa, in A549 and stronger in RERF cells. The intensity of both 130 kDa and 105 kDa bands detected by EP1808Y decreased after 25nM siNrf2 and in A549 Nrf2 KO cells (Figure 1), meaning they are forms of Nrf2 protein.

Next, we exposed cells to the translation elongation inhibitors, to see how stable the Nrf2 forms are. We used cycloheximide (CHX) and emetine at indicated time points and detected Nrf2 (Figure 3). Despite Nrf2 is considered a labile protein of a half-life ranging from less than 30 min~2 h depending on the cell type [5,27,28,29], the ~105 kDa form detected by Abcam EP1808Y antibodies was very stable and still observed even after 24 h of elongation inhibition in both cell lines. The ~130 kDa Nrf2 form detected by both EP1808Y and D1Z9C antibodies was notably less stable and not detected after 8 h chx/emetine treatment in RERF cells. In A549 cells, the ~130 kDa Nrf2 was more stable than in RERF cells, which is consistent with an aberrant Nrf2 degradation in these cells. 

In the next step we studied the cellular distribution of Nrf2 forms. We performed nuclear and cytoplasmic fractionation of A549 and RERF cells and analysed the distribution of Nrf2 under no stress conditions with D1Z9C and EP1808Y antibodies (Figure 4). The ~130 kDa Nrf2 form was detected by both antibodies only in A549 cells and prevalently accumulated in the nucleus, consistent with the constitutive activation of Nrf2 in these cells. Interestingly, the 105 kDa Nrf2 form detected in both cell lines only by EP1808Y antibodies also accumulated in the nucleus. The *in situ* Nrf2 localization with the immunofluorescence confirmed the fractionation results (Figure 5). EP1808Y antibodies detected nuclear and cytoplasmic Nrf2 in both cell lines while D1Z9C recognized primarily nuclear Nrf2.

Since the stable 105 kDa Nrf2 form translocates to the nucleus, in the next step we aimed to check if it is also regulated through the Keap1-Cul3-dependent mechanism. We made use of MLN4924 neddylation inhibitor, a specific small molecule inhibitor of NEDD8-activating enzyme E1 (NAE), that catalyzes the addition of a ubiquitin-like protein NEDD8 (Neural precursor cell expressed developmentally down-regulated 8) to cullins. Neddylation is necessary for the full activation of the Cullin-Ring ligases (CRLs) [30]. MLN4924 binds to the NAE and blocks its enzymatic activity. Consequently, it inhibits the neddylation of all cullins, leading to the accumulation of their substrates [31,32]. Since the Keap1-dependent regulation of Nrf2 requires active Keap1-Cul3-E3 ligase complex, MLN4924 leads to the inhibition of neddylation and disability of Cul3-containing E3 ubiquitin ligase to target Nrf2 for ubiquitination and degradation by proteasome. Consequently, Nrf2 accumulates after MLN4924 treatment, as shown in Figure 6. However, only the ~130 kDa Nrf2 form was accumulated after MLN4924, while the 105 kDa remained unaffected by neddylation inhibition. This indicates that the 105 kDa Nrf2 form is regulated independently of Keap1-Cul3-E3 ligase system. 

The question that arises here, is what is the source of these two distinct Nrf2 forms. One option is the post-translational modifications of Nrf2 that could account for differences in mass and stability of the two forms. Phosphorylation is the predominant Nrf2 modification and various Nrf2 residues are phosphorylated, including Ser40, Ser215, Ser344, Ser347, Ser408, Ser558, Thr559, and Tyr576 (Figure 1) [33,34,35,36,37]. Their impact on Nrf2 stability and activity can be different and is thought to depend on the phosphorylation site. Thus we have asked if two Nrf2 forms detected with Abcam EP1808Y antibodies are the phosphorylated and dephosphorylated Nrf2 forms. We made use of lambda protein phosphatase (λPP), which removes phosphate groups from phosphorylated serine, threonine and tyrosine residues and observed that in both A549 and RERF cells the heavier upper band is affected by λPP treatment, but the lower 105 kDa band is not (Figure 7). Interestingly, after λ phosphatase, the molecular weight of the upper Nrf2 form was reduced to ~110 kDa, which is visible as a band migrating just above the abundant 105 kDa Nrf2. It indicates that the 105 kDa Nrf2 is not simply a dephosphorylated Nrf2, but rather a shorter Nrf2 form.

Our observations that the 105 kDa Nrf2 form: (1) is unexpectedly stable, (2) is not regulated via Keap1-Cul3 ubiquitin ligase 3) migrates faster than the dephosphorylated Nrf2, indicate that it might represent a shorter Nrf2 form that originates from an alternative transcription or/and translation and, most probably, has a disturbed expression or availability of the Keap1-binding motifs: DLG and ETGE. The skipping of exon 2 that encodes these motifs or exons 2+3 in *NFE2L2* gene was observed in lung and head and neck cancers, resulting in Nrf2 forms resistant to Keap1-mediated degradation [15]. To analyze the *NFE2L2* transcripts expressed in A549 cells we have studied the RNA sequencing data and identified six different transcripts expressed under homeostatic conditions in these cells (Figure 7). One of these transcripts, NM_001313904.1, has an extremely short exon 2, translated to three amino acids, due to an alternative translation initiation site and a truncated 3’terminus of this exon (in-frame splice site in the 3’ region of exon 2) (Figure 8A). Nrf2 expressed from this transcript lacks Keap-1 binding regions thus, most probably, escapes Keap1-Cul3 degradation. Transcript NM_001313902.1 on the other hand has a full sequence of exon 2 but lacks exon 3, due to the alternative splicing (Figure 8B). The protein expressed from this transcript might fold in a way that DLG and ETGE motifs are not accessible to the Keap1. Both of these transcripts could be the potential source of the 105 kDa Nrf2 form. It is difficult to hypothesize which transcript it is, basing on the calculated molecular weight (MW) of the Nrf2 form it encodes, as Nrf2 migrates in SDS-PAGE much slower than the calculations suggest. The transcript NM_001313904.1 with alternative translation initiation site encodes for a protein of a calculated MW of 56 kDa (protein isoform 6, NCBI Reference Sequence: NP_001300833.1), while the Nrf2 form encoded by transcript NM_001313902.1 (protein isoform 4, NP_001300831.1) calculates at 64.5 kDa. Since the predicted MW of the full-length *NFE2L2* transcript (NM_006164.5 encoding isoform 1, NP_006155.2) is 68 kDa and all the isoforms migrate in 8% SDS-PAGE above 100 kDa, further studies are needed to reliably assess the origin of the 105 kDa Nrf2 form identified in this study. 

## 4. Discussion

The canonical Nrf2 regulation assumes its constant synthesis and, when there is no stress, a constitutive degradation via Keap1-Cullin3 ubiquitin ligase complex. Accordingly, Nrf2 activation strictly depends on its de-repression via stress-induced modifications of the “stress sensor”—Keap1 [3,38,39]. Indeed, targeted deletion of the *KEAP1* gene in mice resulted in a constitutive accumulation of Nrf2 protein in the nucleus and expression of Nrf2 target genes [40]. Though the Nrf2 expression and function are controlled on various levels (thoroughly reviewed in [41]), it is not clear how Nrf2 maintains the basal expression of its target genes under homeostatic conditions. It has been suggested that Nrf2 is primarily a nuclear protein and the Keap1-Cul3-mediated degradation takes place downstream the transcriptional activity [42,43], though such a model requires Keap1 shuttling from nucleus to cytoplasm that some claim not to observe [44].

Here we found that a stable and a shorter form of Nrf2 exists, that is primarily recognized by EP1808Y Abcam antibodies. This form is not regulated by Keap1-Cul3 complex, nor by any other cullin-RING ubiquitin ligases since they all need to be neddylated for activation. Inhibition of neddylation led to the upregulation of the 130 kDa, but not the 105 kDa Nrf2 form, meaning that the 105 kDa Nrf2 is not classically regulated. Moreover, this form is similarly abundant both in cells with functional Keap1-Nrf2 pathway like RERF cells and in A549 cells, with impaired Nrf2 degradation (due to *KEAP1* mutation). The 130 kDa form, detected by both EP1808Y and D1Z9C antibodies, was found to be a phosphorylated form of Nrf2 thus it was more abundant in A549 cells where Nrf2 is constitutively active, than in RERF cells with wt Keap1, under homeostatic conditions. Upon phosphatase treatment, it was reduced to ~110 kDa, meaning that the 105 kDa form is not simply the dephosphorylated version of ~130 kDa Nrf2. Interestingly, the 105 kDa Nrf2 was resistant to phosphatase treatment, though both 130 kDa and 105 kDa forms were detected in the nucleus indicating they both might be transcriptionally active. It is thus quite likely that the 105 kDa form is a shorter, truncated version of the full length Nrf2 that, due to its truncation, is unable to interact with Keap1 and undergo Cul3-mediated degradation. Since this form is expressed in both cell lines on comparable levels, it might be responsible for regulating basal expression of Nrf2 genes.

It is possible that the 105 kDa form originates from an alternative transcription or/and translation event. Among six *NFE2L2* transcripts expressed in A549 cells under no stress conditions, one is missing almost the whole exon 2, due to an alternative translation initiation and splicing of the 3’ end of this exon (NM_001313904.1), and the other has exon 3 spliced out (NM_001313902.1). Nrf2 produced from the first transcript lacks the Keap1 interactive motifs and the product of the second) transcript could produce a form in which DLG and ETGE regions are less accessible for Keap1, though this awaits experimental verification. Skipping of exon 2 and both exon 2 and 3 of *NFE2L2* gene was shown to occur in lung and head and neck cancers to produce constitutively active Nrf2 forms resistant to Keap1-mediated degradation [15]. It is thus probable that a similar mechanism is utilized to assure the basal expression of Nrf2 targets. As Nrf2 regulates not only stress response, but also physiological cellular processes, expression of a stable Nrf2 form would assure that under homeostatic conditions the Nrf2 program is still active. How such a form would be regulated is another question.

Interestingly, the 105 kDa Nrf2 signal detected by EP1808Y antibodies was before claimed to be unspecific and most probably originating from a protein co-migrating with Nrf2 in HepG2 cells [11]. This conclusion was drawn since authors did not manage to decrease this signal with siNrf2 RNAs nor by depleting the protein with other anti-Nrf2 antibodies. We suspect that difficulties in knocking down the 105 kDa form result from its exceptional stability. In our hands, the use of mixture of different Nrf2-targeting siRNAs and a low number of cells at the time of transfection allowed for a successful reduction of the 105 kDa form expression. Importantly, this form was also significantly reduced in the functional Nrf2 knockout cells.

Another interesting aspect is that, while Abcam EP1808Y antibodies detect two Nrf2 forms, Cell Signaling D1Z9C recognize primarily the 130 kDa Nrf2 with one exception, which is Figure 1, where they detected the 105 kDa Nrf2. As we later show, the 130 kDa Nrf2 is the phosphorylated Nrf2 form. Phosphorylation of Nrf2 is induced by various stimuli, including cold [12]. In all the experiments except from Figure 1, cell pellets were snap frozen before Western blot analysis, while in experiment presented in Figure 1, cells were lysed without a prior freezing. It is possible that freezing induced Nrf2 phosphorylation, and the phosphorylated Nrf2 form was primarily recognized by D1Z9C antibodies. When Nrf2 was not phosphorylated, D1Z9C antibodies detected the 105 kDa form. In A549 cells, Nrf2 is constitutively active and phosphorylated, thus the 130 kDa form is detected regardless the samples preparation conditions.

## Figures and Tables

**Figure 1 antioxidants-10-00786-f001:**
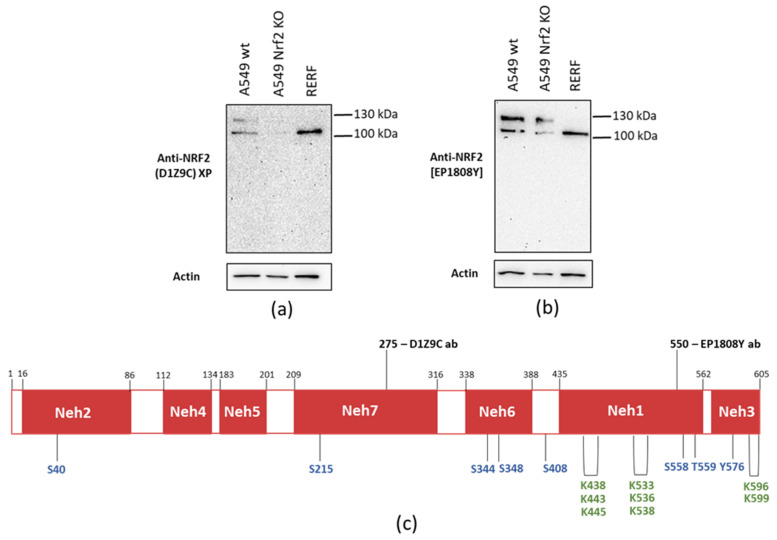
Nrf2 migratory pattern detected in A549 wt, A549 Nrf2 KO, and RERF cells with two monoclonal antibodies—Cell Signaling (CS) (D1Z9C) (**a**) and Abcam [EP1808Y] (**b**), (**c**) Domain structure of Nrf2 with post-translational modifications and regions recognized by Cell Signaling (CS) (D1Z9C) and Abcam [EP1808Y] antibodies. Precise epitope sequence is disclosed, for Abcam EP1808Y it is known to be localized in the C-terminus, surrounding 550 aa and for CS—in the proximity of 275 Ala. Post-translational modifications of Nrf2 residues are marked—phosphorylation in blue (Ser40, Ser215, Ser344, Ser348, Ser408, Ser558, Thr559, and Tyr576) and acetylation in green (Lys438, Lys443, Lys445, Lys533, Lys536, Lys538, Lys596, and Lys599).

**Figure 2 antioxidants-10-00786-f002:**
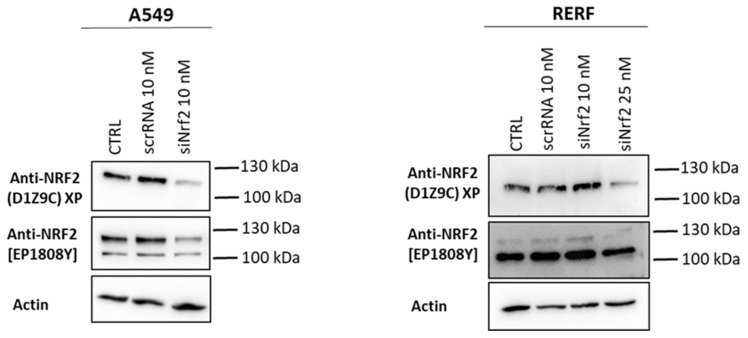
Nrf2 knockdown in A549 and RERF cells detected by two different monoclonal antibodies. Nrf2 expression was silenced with Nrf2-targeting siRNAs for 48 h: 10 nM (A549) and 10 nM and 25 nM (RERF). Nrf2 signal was detected with two monoclonal antibodies: Cell Signaling (D1Z9C) and Abcam [EP1808Y]. Actin was used as a loading control. scrRNA is a control unspecific siRNA.

**Figure 3 antioxidants-10-00786-f003:**
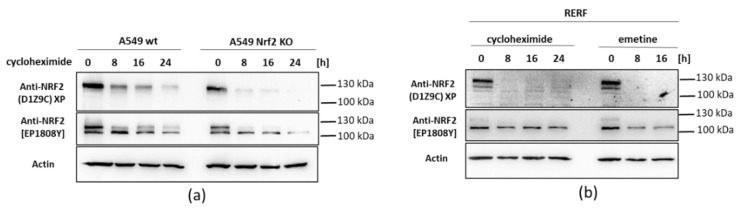
Nrf2 after translation inhibition. (**a**) Western blot analysis of A549 wt and A549 Nrf2 KO cells after treatment with translation inhibitors cycloheximide (chx) at different time points (8, 16, and 24 h). Nrf2 was detected using Cell Signaling (D1Z9C) and Abcam [EP1808Y] antibodies. (**b**) Western blot analysis of RERF cells after treatment with translation inhibitors cycloheximide (chx) and emetine at different time points (8, 16, and 24 h). Nrf2 was detected using Cell Signaling (D1Z9C) and Abcam [EP1808Y] antibodies.

**Figure 4 antioxidants-10-00786-f004:**
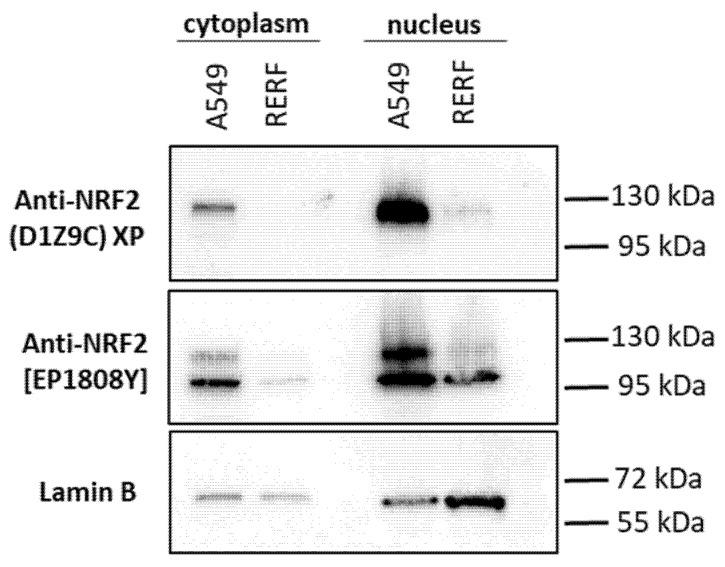
Western blot analysis after nuclear and cytoplasmic fractionation of A549 and RERF cells. Nrf2 was detected with two monoclonal antibodies: Cell Signaling (D1Z9C) and Abcam (EP1808Y) rabbit. Lamin B was used as a nuclear marker showing enrichment in the nuclear fraction.

**Figure 5 antioxidants-10-00786-f005:**
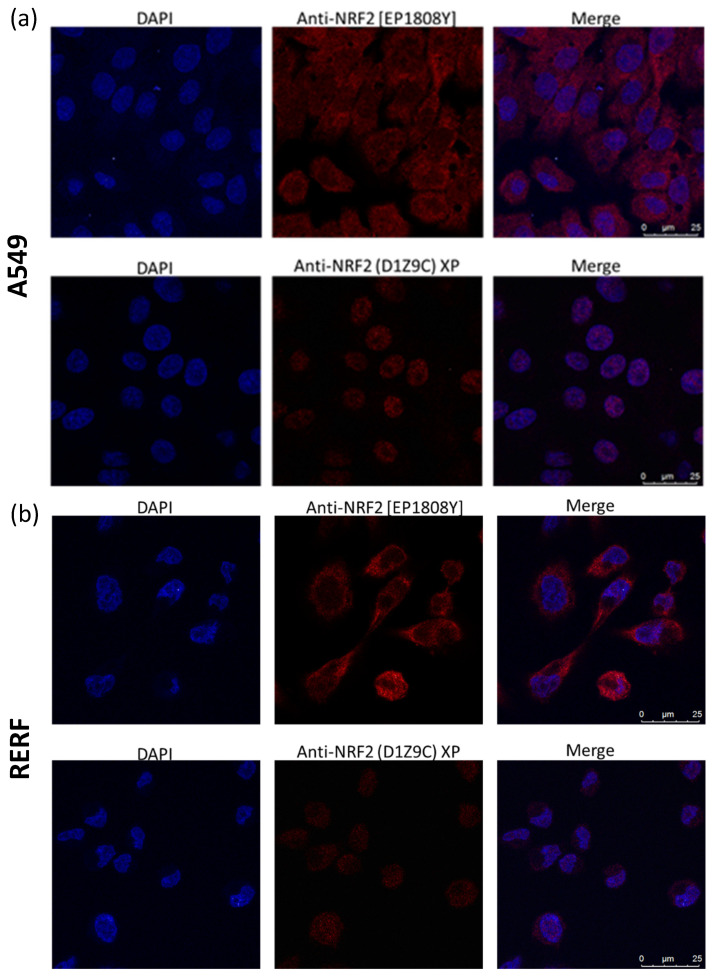
Cellular Nrf2 localization detected with Nrf2-specific monoclonal antibodies EP1808Y and D1Z9C in A549 (**a**) and RERF (**b**) cells. After fixation, cells were stained with Abcam (EP1808Y) and Cell Signaling (D1Z9C) antibodies followed by Alexa Fluor 488 goat anti-rabbit secondary antibodies. Nuclei were stained with DAPI.

**Figure 6 antioxidants-10-00786-f006:**
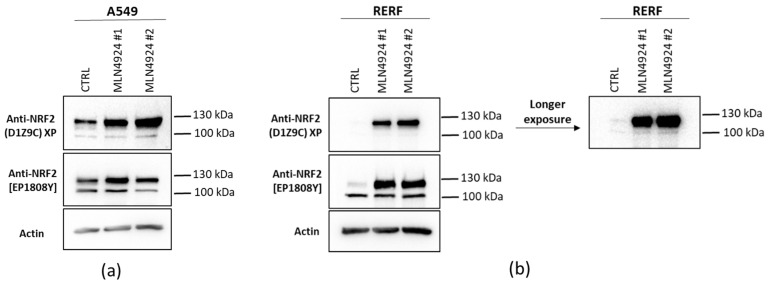
Western blot analysis of Nrf2 in A549 wt (**a**) and RERF (**b**) cell lines, after treatment with neddylation inhibitor MLN4924. Cells were treated with 1 μM MLN4924 for 12 h. Nrf2 was detected using indicated antibodies. #1, #2 represent two repetitions of MLN4924 treatment. Actin was used as a loading control.

**Figure 7 antioxidants-10-00786-f007:**
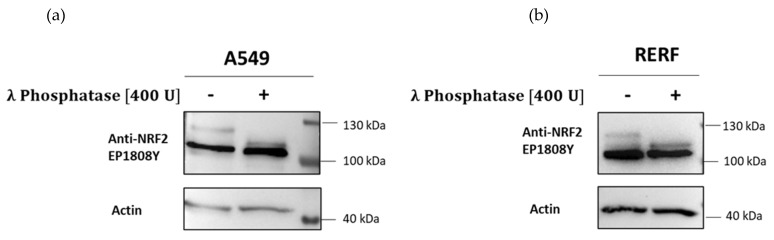
Lambda protein phosphatase (λPP) treatment of A549 (**a**) and RERF (**b**) lysates. Cell lysates were incubated with or without λ phosphatase for 30 min in 30 °C in the presence of MnCl_2_ and Nrf2 was detected by Western blot with Abcam EP1808Y antibodies.

**Figure 8 antioxidants-10-00786-f008:**
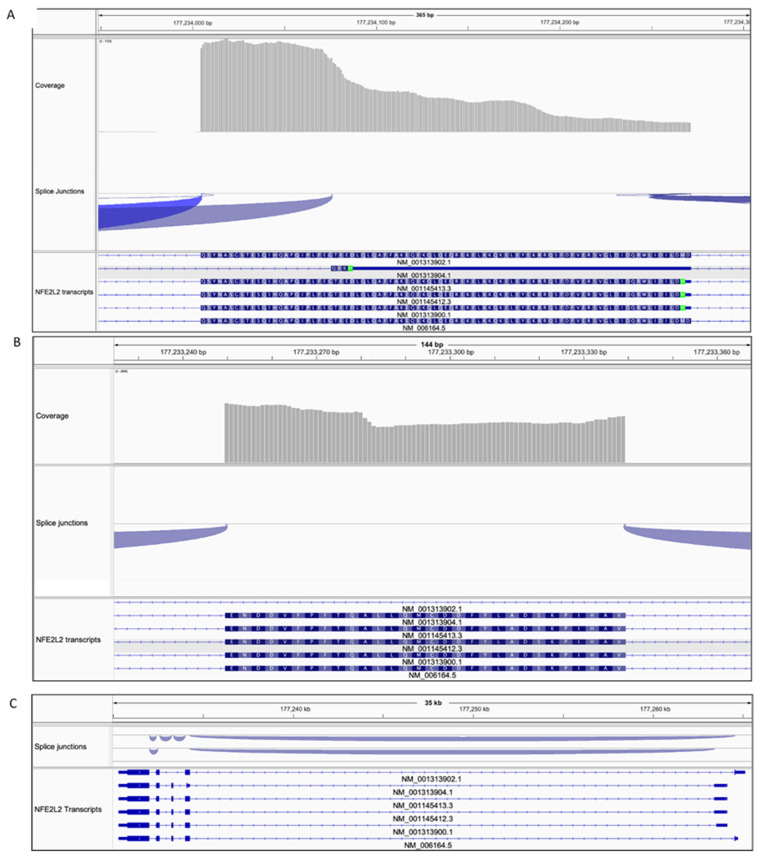
Expression of different *NFE2L2* transcripts in A549 cells based on RNA sequencing data. (**A**) Visualization of sequence reads alignments within exon 2. Beginning of the exon 2 is on the right-hand and the end on the left-hand side. Splice junctions are represented by arches. Apart from classical junctions at the beginning and the end of the exon, in a portion of reads the 3’ end of exon 2 was spliced out. These reads have been aligned to the transcript NM_001313904.1 (transcripts references are below each transcript). Transcripts sequence within the exon was translated to the encoded amino acids (**B**) Visualization of sequence reads alignments within exon 3. In the transcript NM_001313902.1 exon 3 is spliced out. (**C**) Visualization of all the transcripts (6) identified in A549 cells. Exon 1 is on the right-hand and exon 5 on the left-hand side. Assembly of RNA-Seq alignments into potential transcripts together with calculation of expression levels for those transcripts were performed with StringTie software [20]. Visualization of the alignments identified transcripts and junctions was performed in IGV software [21].

## Data Availability

Data is contained within the article. The file with the original western blot images (whole membranes) presented in this article has been submitted to the journal together with the article.

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
