# Peer review of "Identification of a Stable, Non-Canonically Regulated Nrf2 Form in Lung Cancer Cells"

_antioxidants, 2021, doi:10.3390/antiox10050786_

Round 1
Reviewer 1 Report
I found this paper very interesting, since nrf2 is my field ofinterest and finally this study clarifies the apparently anomalous
electrophoretic pattern of Nrf2. We hope that the knowledge that accumulates
on the regulation of this pathway will lead to advances in the field of
cancer treatment and chronic inflammatory diseases.
Author Response
We would like to thank the Reviewer for the appreciation of this study. In the revised version we include the lambda phosphatase treatment revealing that 130 kDa Nrf2 is a phosphorylated Nrf2 form and the analysis of RNA-sequencing data indicating that the stable 105 kDa Nrf2 might originate from an alternative splicing of exon 2 or exon 3.
Reviewer 2 Report
The premise of this work boils down to evaluating two different commercial antibodies that recognize Nrf2. While it is refreshing to see a stringent analysis of antibody quality, it is unfortunate that this was not done in the context of a significant biological question. That said, the study was generally well executed and provides valuable information for the field. There are a few suggestions for improving clarity and quality:
- The rationale for the study is weak. As stated it is "...discrepancies concerning migratory pattern using different antibodies has cast a cloud of uncertainty over some studies." The work presented would be significantly strengthened by more specific details about the discrepancies and clear indication that these are novel findings.
- Figure 1 should include locations of neddylation and other known modification sites
- It seems odd that a completely null cell line wasn't use to unequivocally establish specificity...is that line not viable?
- The authors refer to a "95 kda" band, but it is clearly larger...actual measurement of apparent MW relative to markers would be appropriate in a study like this.
- In Figure 1 the D1Z9C antibody dectects both bands and primarily the lower band. For the rest of the paper this antibody is reported to pretty much exclusively detect the larger band. If there were condition changes that were responsible, then they should be stated.
- For the CHX assay, this should not be described as a "chase" assay - at least not as described. A chase assay is the latter half of a pulse-chase where a label or drug is chased with unlabeled compound or a wash. This looks like continuous exposure to CHX or emetine.
- Figure 5 panel b shows a control lane with no signal, but all other blots with this condition showed clear signal...this is a major discrepancy from prior figures.
- Seems odd to have a single supplementary figure - I think it should be included in the main text. However, Regarding glycosylation, what reason do the authros think it may be glycosylated - especially by the N-linked moieties that are sensitive to PNGase F. If there is prior work on what type of glycosylation is present, it would be good to cite.
- The mechanism figure (6) is problematic for several reasons. First, why change the cell type. Second overexpression is probably the least likely method to reveal mechanism. It would be much more informative to take a bioinformatic approach. NCBI says there are 5 isoforms of Nrf2 transcripts that result in different length proteins....do any of these map to the sizes seen? Are they missing known sites of modification?
- This study highlights the value of Northern blots which seem to be a dead art. In a single experiment the authors could visualize the relative abundance of transcripts and determine if any might correspond to the sizes observed.
- Regarding English language, the quality is OK, but lines 231-240 exemplify an overall lack of clear writing style that could use some polishing.
Round 2
Reviewer 2 Report
No further comments